# Sleep quality and its correlates among undergraduate medical students in Nepal: A cross-sectional study

Kiran Paudel[1,2]*, Tara Ballav Adhikari[2,3,4], Pratik Khanal[1], Ramesh Bhatta[5], Rajan Paudel[1], Sandesh Bhusal[1,2], Prem Basel[1]

1 Institute of Medicine, Tribhuvan University, Maharajgunj, Kathmandu, Nepal, 2 NCD Watch Nepal, Maharajgunj, Kathmandu, Nepal, 3 Department of Public Health, Section for Global Health, Aarhus University, Aarhus, Denmark, 4 Nepal Development Society, Chitwan, Nepal, 5 Asian College for Advanced Studies, Purbanchal University, Satdobato, Lalitpur, Nepal

* kiranpaudel59@gmail.com

## Abstract

Poor sleep quality has been found to affect students' learning abilities, academic performance, and interpersonal relationships. However, little is known about this issue in Nepal. This study aimed to identify the factors associated with poor sleep quality among undergraduate medical students in Nepal. A web-based survey was conducted in March 2021 among 212 undergraduate medical students at the Institute of Medicine, Kathmandu, Nepal. Sleep quality was measured using a 19-item Pittsburgh Sleep Quality Index (PSQI). Multivariable logistic regression analysis was done to assess the factors associated with sleep quality. In the study participants, 38.2% of the students were identified as poor sleepers. Factors like being depressed (AOR = 4.5, 95% CI; 1.2–5.4), current alcohol use (AOR = 2.5, 95% CI; 1.8–10.8), poor academic achievement (AOR = 3.4, 95% CI; 1.1–10.9), and being a fourth-year student (AOR = 3.6, 95% CI; 1.1–11.8) were significantly associated with poor sleep quality. Poor sleep quality was common among undergraduate medical students. Routine screening of sleep quality and depressive symptoms is necessary to mitigate their impact among medical students. Medical students of the fourth year, current alcohol users, and those who did not have good academic achievement had poor sleep quality. Special attention on these population subgroups is thus needed to enhance sleep quality.

## Introduction

Sleep quality is one's satisfaction with sleep experience, integrating aspects of sleep initiation, sleep maintenance, sleep quantity, and refreshment upon awakening [1]. Sleep is a basic human need of every person's overall health and wellbeing, which is affected by various factors such as physical, mental, and environmental [2]. Epidemiological evidence suggests that sleep duration and poor sleep are associated with premature mortality and various adverse health outcomes like cardiovascular diseases, immune system suppression, obesity, migraine, etc. [3,4]. A global review revealed that sleep disturbances affect an important proportion of medical students ranging from 41% of the participating students in Iran, 70% in Hong Kong, to

**Data Availability Statement:** The data used for the analysis is included in the Supporting information files.

**Funding:** This study was funded by undergraduate research grant from the Nepal Health Research Council (NHRC), Government of Nepal under Undergraduate Health Research Grant Program 2021 (Ref no. 2153). Kiran Paudel (KP) was the recipient of this grant. The funders had no role in study design, data collection and analysis, decision to publish, or preparation of the manuscript.

**Competing interests:** The authors have declared that no competing interests exist.

90% in China [5]. Studies suggested that 50 to 70 million American people were chronically suffering from sleep disorders [6,7]. Proper sleep aids in the optimum functioning of the brain, which consequently helps to improve knowledge and grasp new concepts [8]. The problem of poor sleep quality is faced by university students where academic demand is fairly high [9].

Medical students are prone to stress because of their highly demanding professional roles and academic requirements [10]. Several studies have demonstrated that university-level students from different countries, like 24% from the United Kingdom, 30% in Korea, and 49% in Taiwan, slept less than 7 hours per night [11–13]. The academic performance of many students is affected by their inadequate sleeping habits, which is not much realized by students [14]. Studies in developing countries revealed that 32.5–76% of medical students suffer from poor sleep quality [15,16]. In previous studies conducted in Nepal, the prevalence of poor sleep quality was reported to be 44.2%, 30.3% among medical students, and 35.4% among undergraduate non-medical students [17–19].

A higher prevalence of poor sleep quality among medical students than non-medical students and the general population has also been reported [5]. Various factors, including medical students' attitudes, knowledge of sleep, and academic demands, have been recognized as the causative factors [5]. Sleep deprivation can lead to depression, suicide, and a high chance of substance abuse among adolescence [20]. Despite inadequate vacation and long duration of hospital posting along with academic sessions and stressful lifestyles, there is limited evidence regarding the burden of poor sleep quality among medical students in Nepal. Our study in a public funded medical college is first of its kind as previous published studies were conducted in private medical colleges. Students with the highest academic standing are considered to study at publicly funded medical colleges. Due to the high flow of patients, students have to perform more clinical duties. Therefore, this study aimed to determine the prevalence of poor sleep quality and its correlates among medical students at the undergraduate level in a medical college in Nepal.

## Methods

### Study design and setting

A cross-sectional study was conducted in the Maharajgunj Medical Campus (MMC), Institute of Medicine (IOM), under Tribhuvan University in March 2021. It is the largest government medical college in Nepal and receives students from different parts of the country. It offers medical, nursing, public health, and other health sciences courses at undergraduate, post-graduate, and super-specialty levels. Bachelor of Medicine and Bachelor of Surgery (MBBS) degree program is a 5.5 year-long course divided into preclinical and clinical stages. In 2020/21, a total of 383 students were studying MBBS at this campus.

### Study population

The study participants were selected randomly from all undergraduate medical students (year one up to final year proportionately) studying at MMC. Selected participants who did not respond to our request could not be included in the study. There could be multiple reasons for this: not having internet access, lacking interest in participating, or not checking their emails. Those who did not respond after two times follow up were excluded from the study. Also, any students who were below 18 years old were excluded. The study team obtained ethical approval from the Institutional Review Committee of IOM, Tribhuvan University (Ref no. 360-6-11, 2077/2078). The ethics committee approved the use of digital consent. Study objectives were explained in the Google forms, and e-informed consent was taken from all the participants

before the data collection. Study participants participated voluntarily, and students were free to opt-out at any time.

## Sampling

The required number of participants for the study was calculated by using the formula $n = z^2pq/d^2$ with the following assumptions: margin of error 5%, at 95% Confidence Interval (CI), and taking the prevalence of poor sleep quality (p):44.23% [17]. After adding a 30% non-response rate, the final calculated sample size was 250. The list of the students was obtained from the Dean's Office, and participants from each study year were selected using simple random sampling. The list showed that there were 79 students in the first year and 76 each in others years. Random values were obtained for each student using the rand command in Excel, and those with the highest value obtained were selected for participation. The class representative from each study year helped in obtaining email addresses of selected students, and questionnaires were emailed to the study participants individually.

## Study tools

A self-administered questionnaire was distributed to the participants through their email addresses. A reminder was sent through Viber, WhatsApp, and Facebook, wherever possible in case of non-response to the email. The socio-demographic section of study tools consisted of information about participants' sex (male, female), ethnicity (Brahmin/Chettri, Madhesi, Adiwasi/Janjati, Dalit), current living residence (in the hostel, with family, alone), year of the study (first, second, third, fourth, final), family income (less than Nepalese Rupee 20,000, 20,000–50,000, 50,000 and above), academic achievement (pass, fail), and study choice (own preference, family pressure). Assess the students' academic performance, the outcome of the board examination in the previous year was recorded; those who have succeeded were grouped as passed and others as failed. For the first-year students, the outcome of the previous internal examination was recorded. Substance-related factors (current smokers and current alcohol users) were assessed by WHO ASSIST (Alcohol Smoking and Substance Involvement Screening Tool [21]. We grouped students based on their use of tobacco and alcoholic beverages in the past 30 days. Current tobacco users are defined as the students who had consumed a tobacco product either smoking or chewing at least once. Similarly, students who had consumed alcoholic beverages at least once in the past 30 days were defined as current alcohol users.

For the assessment of depression, a nine-item Patient Health Questionnaire-9 (PHQ-9) was used. The tool recorded the frequency of symptoms of depression during the past two weeks. Responses were recorded as "not at all", "several days", "more than half the days" or "nearly every day" and were scored between 0–3. A composite score of 0–27 was generated, and a score $\geq$ 10 was considered a depressive disorder [22].

For the assessment of smartphone addiction, ten items Smartphone Addiction Scale Short Version (SAS-SV) was used. On self-reporting six-point Likert scale (1: "strongly disagree", 2: "disagree", 3: "weakly disagree", 4: "weakly agree", 5: "agree", and 6: "strongly agree"). The scores were summed, and score higher than 32 was considered problematic smartphone addiction [23].

Sleep quality was assessed by using Pittsburgh Sleep Quality Index (PSQI). It differentiated the sleep quality over the past month based on seven domains: subjective sleep quality, sleep latency, sleep duration, habitual sleep efficiency, sleep disturbances, use of sleep medication, and daytime dysfunction. PSQI consists of 19 items and produced a score of 0 (no difficulty) to 3 (severe difficulty) on these seven domains. These domains produced a global score,

ranging from 0 to 21, where a score greater than five was used to determine the poor sleep quality [24].

The tools adapted in this study have been previously used in a similar setting and similar study population in Nepal. For instance, the sleep quality and depression tools were used by Bhandari PM et al. among undergraduate non-medical students in a study published in BMC Psychiatry [19]. Similarly, the same tool of sleep quality was used by Sundas N et al. among undergraduate medical students of a private medical college [17]. Likewise, the tool used to assess smart phone addiction was previously used by Karki S et al. among undergraduate medical students [25]. Inclusion of the individuals independent variables that could affect sleep quality was done by referring to the previous studies [17–19].

### Data analysis

Data from the Google forms were automatically recorded in Google sheets. All the collected information was systematically compiled, coded, checked, and edited on the same day of data collection. Analysis was done using STATA version 15.1 (StataCorp. Texas, USA).

Descriptive analysis of the variables was done in terms of frequency and percentage. The mean score was calculated for Pittsburgh Sleep Quality Index. The Chi-square test was used to determine the association between categorical independent and categorical dependent variables. Variables with a p-value less than 0.1 during bivariate analysis were entered into the regression model [26]. In bi-variable logistic analysis, variables namely depression, smart phone addiction, current alcohol intake status, current smoking status, study year, study choice, academic achievement, and self-reported health problems were found to have a p-value less than 0.1. These variables fulfilled minimum requirements for further multivariable logistic regression and was thus fitted in the final regression model. Multivariable logistic regression analysis was conducted to determine the statistically significant association between explanatory variables and outcome variables. The regression model allowed for the adjustment of multiple variables, thus controlling potentials confounding variables. The adjusted odds ratio was calculated at a 95% CI, and a p-value less than 0.05 was considered statistically significant.

## Results

### Socio-demographic characteristics of study participants

Out of 250 students approached, a total of 212 students participated in the study with a response rate of 85%. Of the study participants, 67.9% were male, 63.2% belonged to the Brahmin/Chhetri ethnic group, and 46.7% lived in the hostel. Nearly half of the participants had a family income of NPR. 20,000–50,000 and 10.8% of the participants studied MBBS due to their family pressure. Most of the participants (89.2%) passed their last academic year exam (Table 1).

### Health and behavioral characteristics of the study participants

Among the study participants, 27.8% had symptoms of depression, and 44.8% of them were addicted to smartphone. Nearly one-fifth of the participants (17.0%) reported having any health problems. The proportion of current smokers and current alcohol users was 26.4% and 42.0%, respectively (Table 2).

### Magnitude of poor sleep quality and its components scores among the respondents

Among study participants, 81 (38.2%) had poor sleep quality. The proportion of poor sleep quality was more female (44.1%) than male (35.4%) participants. Female students had a higher

**Table 1. Socio-demographic characteristics of study participants.**

| Characteristics | Number | Percentage |
|---|---|---|
| **Gender** | | |
| Male | 144 | 67.9 |
| Female | 68 | 32.1 |
| **Ethnicity** | | |
| Brahmin/Chettri | 134 | 63.2 |
| Madhesi | 39 | 18.4 |
| Aadiwasi/Janjati | 27 | 12.7 |
| Dalit | 12 | 5.7 |
| **Current living residence** | | |
| In hostel | 99 | 46.7 |
| With family | 81 | 38.2 |
| Alone | 32 | 15.1 |
| **Family income** | | |
| Less than 20,000 | 21 | 9.9 |
| 20,000–50,000 | 103 | 48.6 |
| 50,000 and above | 88 | 41.5 |
| **Current study year** | | |
| First | 42 | 19.8 |
| Second | 48 | 22.6 |
| Third | 35 | 16.5 |
| Forth | 41 | 19.3 |
| Final | 46 | 21.7 |
| **Academic performance** | | |
| Pass | 185 | 87.3 |
| Fail | 27 | 12.7 |
| **Reason for study choice** | | |
| Own preference | 189 | 89.2 |
| Family Pressure | 23 | 10.8 |
| **Total** | 212 | 100 |

mean global PSQI and a larger portion of poor sleepers (44.1%) than their male counterparts (35.4%). Regarding sleep latency, 5.1% of the participants had sleep latency of more than 60 minutes. One in four study participants reported having bad subjective sleep quality. Four out of five study participants (79.3%) did not use sleep medication in the past one month, and 45 (21.2%) slept for less than six hours in the past one month. There was a significant difference in sleep disturbance status across gender (p-value = 0.01) (Table 3).

## Factors associated with sleep quality

The odds of poor sleep quality among study participants who were depressed were 4.5 (95% CI: 1.8–10.8) times higher than non-depressed respondents. Regarding academic achievement, participants who failed in the last academic year exam were 3.4 (95% CI; 1.1–10.9) times more likely to have poor sleep quality than those who passed. Fourth-year students were 3.6 (95%CI; 1.1–11.5) times more likely to have poor sleep quality than first-year students (Table 4).

## Discussion

This study revealed the prevalence of poor sleep quality among undergraduate medical students (38.2%), which is higher than the study reported among undergraduate non-medical

**Table 2. Health and behavioral characteristics of the study participants (n = 212).**

| Characteristics | Number | Percentage (%) |
|---|---|---|
| **Depression** | | |
| Yes | 59 | 27.8 |
| No | 153 | 72.2 |
| **Smart phone addiction** | | |
| Addicts | 95 | 44.8 |
| Non-addicts | 117 | 55.5 |
| **Currently smoking** | | |
| Yes | 56 | 26.4 |
| No | 156 | 73.6 |
| **Current alcohol users** | | |
| Yes | 89 | 42 |
| No | 123 | 58 |
| **Shared bed with others** | | |
| Yes | 44 | 20.8 |
| No | 168 | 79.2 |
| **Any self-reported health problem** | | |
| Yes | 36 | 17.0 |
| No | 176 | 83.0 |

students (35.4%) [19] and school level students (31%) in Nepal [27]. A similar study from a private medical college of Nepal showed a higher proportion of poor sleep quality (44.3%) than our study [17]. The higher prevalence of poor sleep quality among medical students might be due to the duration of medical education (5.5 years), longer than any other undergraduate courses in Nepal (4 years). Medical students are further vulnerable to academic stress and enjoy less leisure time compared to students of other disciplines [28], which might have aggravated the poor sleeping patterns. The result of the present study is higher than that reported in Nigeria 32% [15] and central India 32.5% [29]. The prevalence of poor sleep quality in this study is comparable with other studies conducted in different countries such as Pakistan (39.5%) [30], and Thailand (42.4%) [31] while lower than in Nigeria 50.1% [32], Sudan 61.4% [33], and Saudi Arab (76%) [16]. The possible reasons for the variability across the countries could be differences in sampling technique, year of medical school, and exposure to a social environment.

The present study demonstrates a significant association between depression and poor sleep quality. Our study finding is compatible with the study conducted among undergraduate students in Nepal [19], Turkey [34], Ethiopia [35], Saudi Arabia [16], and Brazil [36]. A study from Brazil showed that sleep disruption, insomnia, or less than 7 hours of sleep manifests a cumulative risk ratio of 5.5 for minor psychiatric disorders among medical students [20]. Depression decreases serotoninergic neurotransmission levels, which may affect regular sleep patterns [37]. The study finding is important for paying attention to the depression and sleep problems of the medical students that they experience during their medical school. It is critical that the current teaching-learning environment at medical school be assessed before recommending any interventions for enhancing medical education and improving the well-being of medical students.

We also found alcohol consumption associated with poor sleep quality. The finding of this study coincides with the study conducted in Korea, which showed poor subjective sleep quality among alcohol consumers [38]. Drinking alcohol can cause sleep disorders because it disrupts

**Table 3. Sleep quality and sleep patterns overall stratified by gender.**

| Characteristics (n = 212) | Female (%) | Male (%) | All (%) | $\chi^2$ value | p-value |
|---|---|---|---|---|---|
| **Overall sleep quality** | | | | | |
| **Mean PSQI (±SD)** | 5.91±3.5 | 5.1±3.2 | 5.36±3.3 | | 0.1[#] |
| **Sleep quality** | | | | | |
| Good | 38 (55.9) | 93 (64.6) | 131 (61.8) | 1.5 | 0.2 |
| Poor | 30 (44.1) | 51 (35.4) | 81 (38.2) | | |
| **Subjective sleep quality** | | | | | |
| Very good | 19 (27.9) | 48 (33.3) | 67 (31.6) | 2.3 | 0.5 |
| Fairly good | 30 (44.1) | 62 (43.1) | 92 (43.4) | | |
| Fairly bad | 16 (23.5) | 32 (22.2) | 48 (22.6) | | |
| Very bad | 3 (4.4) | 2 (1.4) | 5 (2.4) | | |
| **Sleep latency** | | | | | |
| ≤15 minutes | 19 (27.9) | 53 (36.8) | 72 (34.0) | 1.8 | 0.6 |
| 16–30 minutes | 31 (45.6) | 55 (38.2) | 86 (40.6) | | |
| 31–60 minutes | 14 (20.6) | 29 (20.1) | 43 (20.3) | | |
| ≥60 minutes | 4 (5.9) | 7 (4.9) | 11 (5.1) | | |
| **Sleep duration** | | | | | |
| >7 hours | 24 (35.3) | 52 (36.1) | 76 (35.9) | 6.6 | 0.08 |
| 6–7 hours | 29 (42.7) | 62 (43.1) | 91 (42.9) | | |
| 5–6 hours | 12 (17.6) | 30 (20.8) | 42 (19.8) | | |
| <5 hours | 3 (4.4) | 0 | 3 (1.4) | | |
| **Sleep efficiency** | | | | | |
| >85% | 45 (66.2) | 113 (78.5) | 158 (74.5) | 5.5 | 0.1 |
| 75–84% | 17 (25) | 18 (12.5) | 35 (16.5) | | |
| 65–74% | 4 (5.9) | 10 (6.9) | 14 (6.6) | | |
| <65% | 2 (2.9) | 3 (2.1) | 5 (2.4) | | |
| **Sleep disturbance** | | | | | |
| 0 (Not during the past month) | 4 (5.9) | 32 (22.2) | 36 (17) | 9.9 | **0.01** |
| 1 (Less than once a week) | 50 (73.5) | 84 (58.3) | 134 (63.2) | | |
| 2 (Once or twice a week) | 9 (13.2) | 22 (15.3) | 31 (14.6) | | |
| 3 (Thrice or more than a week) | 5 (7.4) | 6 (5.2) | 11 (5.2) | | |
| **Use of sleep medication** | | | | | |
| Not during the past month | 53 (78) | 115 (79.9) | 168 (79.3) | 1.0 | 0.8 |
| Less than once a week | 6 (8.8) | 14 (9.7) | 20 (9.4) | | |
| Once or twice a week | 6 (8.8 | 12 (8.3) | 18 (8.5) | | |
| Thrice or more than a week | 3 (4.4) | 3 (2.1) | 6 (2.8) | | |
| **Day time dysfunction** | | | | | |
| Never | 31 (45.6) | 73 (50.7) | 104 (49) | 3.2 | 0.4 |
| < once a week | 23 (33.8) | 41 (28.5) | 64 (30.2) | | |
| 1–2 times per week | 8 (11.8) | 24 (16.7) | 32 (15.1) | | |
| ≥3 times per week | 6 (8.8) | 6 (4.1) | 12 (5.7) | | |

[#]independent t-test.

the sequence and duration of the sleep period [39]. Consumption of alcohol causes an imbalance of the sleep homeostatic mechanism of the body [40]. In both ways, physically or mentally and physiologically, alcohol consumption might impact sleep quality. Activities focusing on reducing alcohol use should thus be a core agenda for improving sleep quality among medical students.

**Table 4. Logistic regression analysis showing association between explanatory variables and sleep quality.**

| Explanatory variables | Sleep quality N (%) | | Crude Odds ratio (95% CI) | Adjusted Odds ratio (95% CI) |
|---|---|---|---|---|
| | Good (n = 131) | Poor (n = 81) | | |
| **Depression** | | | | |
| Yes | 18 (30.5) | 41 (69.5) | 6.4 (2.2–12.5) | 4.5 (1.8–10.8) ** |
| No | 113 (78.9) | 40 (26.1) | Ref | Ref |
| **Smart phone addiction** | | | | |
| Addicts | 43 (45.3) | 52 (54.7) | 3.7 (2.0–6.5) | 1.7 (0.8–3.6) |
| Non-addicts | 88 (75.2) | 29 (24.8) | Ref | Ref |
| **Current alcohol intake status** | | | | |
| Yes | 37 (41.6) | 52 (58.4) | 4.6 (2.5–8.2) | 2.5 (1.2–5.4) * |
| No | 94 (76.4) | 29 (23.6) | Ref | Ref |
| **Current smoking status** | | | | |
| Yes | 22 (39.3) | 34 (60.7) | 3.6 (1.9–6.8) | 0.9 (0.4–2.2) |
| No | 109 (69.9) | 47 (40.1) | Ref | Ref |
| **Study year** | | | | |
| First | 35 (83.3) | 7 (16.7) | Ref | Ref |
| Second | 28 (58.3) | 20 (41.7) | 3.6 (1.3–9.6) | 1.3 (0.4–4.2) |
| Third | 24 (68.6) | 11 (31.4) | 2.3 (0.8–6.7) | 1.5 (0.4–5.1) |
| Forth | 19 (46.3) | 22 (53.7) | 5.8 (2.1–16.0) | 3.6 (1.1–11.5)* |
| Final | 25 (54.3) | 21 (45.7) | 4.2 (1.5–11.4) | 1.8 (0.5–5.9) |
| **Shared bed with others** | | | | |
| Yes | 23 (52.3) | 21 (47.7) | 1.6 (0.8–3.2) | 1.2 (0.5–2.8) |
| No | 108 (64.3) | 60 (35.7) | Ref | Ref |
| **Study Choice** | | | | |
| Family pressure | 10 (43.4) | 13 (56.1) | 2.3 (0.9–5.5) | 0.9 (0.3–3.0) |
| Own preference | 121 (64.1) | 68 (35.9) | Ref | Ref |
| **Academic achievement** | | | | |
| Fail | 8 (29.6) | 19 (70.4) | 4.8 (1.9–11.3) | 3.4 (1.1–10.9)* |
| Pass | 123 (66.5) | 62 (33.5) | Ref | Ref |
| **Any self-reported health problem** | | | | |
| Yes | 10 (27.8) | 26 (72.2) | 5.7 (2.6–12.7) | 1.4 (0.5–3.9) |
| No | 121 (68.8) | 55 (31.3) | Ref | Ref |

* $p$-value less than 0.05

** $p$-value less than 0.001.

In this study, the odds of poor sleep quality were three times higher among participants studying in the fourth year than in the first year. Similar findings were reported from the study in Nepal [17] and Brazil [36]. A previous study conducted among medical students in Nepal showed a high prevalence of excessive daytime sleepiness [41] and prone to poor sleep quality. Students in the fourth year have higher pressure due to increased academic schedule and workload as they progress from pre-clinical to clinical year [42]. The possible reasons might be the high number of lectures, difficulty in handling the study pressure, clinical rotations in the hospital, and fewer academic breaks. According to the curriculum, students in the fourth year have to face a series of examinations, putting students under tough mental pressure. Additionally, failure in the fourth year leads to the discontinuation of hospital-based internships. However, students in the fifth year need not face examinations which might have contributed to such differing results.

Student's academic success also affected sleep quality among medical students. Failure in the previous board exam's result or internal exam for first-year students was associated with higher odds of poor sleep quality. This finding is consistent with the previous studies conducted in Saudi Arabia [16], China [43], and Ethiopia [44]. A study conducted among medical students at the College of Medicine, King Saud University, showed an association between poor sleep quality and poor concentration. Furthermore, sleep disturbed students are usually unaware that sleep deprivation can negatively impact their examination preparation and performance and impair their ability to complete the task [45,46]. Similarly, those students who failed might have spent more time reading at night, depriving themselves of proper sleep. Interestingly, we found no significant association of sleep quality with current smoking and smartphone addiction, although studies done elsewhere have shown an association of poor sleep quality with these variables [47,48].

The study has some limitations which need to be acknowledged. The risk behaviors and their association being studied cross-sectional; it is impossible to infer directionality in the relationship. There might be respondent bias as the findings were self-reported and based on a subjective scale, resulting in underestimating, and overestimating their behaviors. There might be selection bias that may have occurred due to non-response. Due to selective non-response bias, the prevalence of self-reported variables such as smartphone addiction, alcohol consumption, and tobacco consumption may be significantly underestimated. Students who use tobacco and alcohol may be among those who were not chosen and who did not respond. Similarly, another limitation of this study is that the tools used were not validated in the Nepali language. However, we used questionnaires in the English language that were previously used in similar settings and similar population in Nepal [17–19]. Also, during the pretesting, no one reported the difficulties in understanding the administered questionnaires. The study population were university-level students educating in the English medium.

Despite the limitations, this is the first study from a public medical college in Nepal, which provides evidence on the sleep quality among undergraduate medical students. The study findings could be of particular interest to medical colleges, education experts, and policymakers. The study provides background information on sleep quality to design large-scale studies in different age groups, cognitive and behavioral development, and mental health impacts.

## Conclusion

A high prevalence of poor sleep quality was reported among undergraduate medical students in this study. Depression, alcohol consumption, study year, and academic achievement were significant correlates of poor sleep quality. Based on study findings, we recommend academic counseling focusing on students' mental health status and sleep hygiene and activities focusing on discouraging alcohol consumption. Similarly, redesigning the fourth-year medical syllabus for reducing the academic load is also recommended to reduce academic stress and poor sleep quality. These findings can help inform educators to conduct sleep hygiene promoting programs early within the medical school. Further prospective research would be helpful to investigate the cause-effect relationship of risk factors of poor sleep quality.

## Supporting information

**S1 Data. Data underlying the results of the study.**
(XLSX)

## Acknowledgments

We acknowledge all the study participants for their valuable time and support in completing the study. We would like to thank Shreeyash Bhattarai, Bhoj Raj Kalauni, Biplav Aryal, Durga Rijal, and Prashamsa Bhandari for their support during the data collection process. We appreciate Maharajgunj Medical Campus for providing administrative approval for conducting the study.

## Author Contributions

**Conceptualization:** Kiran Paudel, Tara Ballav Adhikari.

**Data curation:** Kiran Paudel.

**Formal analysis:** Kiran Paudel, Tara Ballav Adhikari.

**Funding acquisition:** Kiran Paudel.

**Investigation:** Kiran Paudel.

**Methodology:** Kiran Paudel, Tara Ballav Adhikari, Rajan Paudel, Sandesh Bhusal.

**Project administration:** Kiran Paudel, Tara Ballav Adhikari.

**Resources:** Kiran Paudel.

**Software:** Kiran Paudel.

**Supervision:** Kiran Paudel, Tara Ballav Adhikari, Pratik Khanal, Prem Basel.

**Validation:** Kiran Paudel, Tara Ballav Adhikari.

**Visualization:** Kiran Paudel.

**Writing – original draft:** Kiran Paudel, Ramesh Bhatta.

**Writing – review & editing:** Kiran Paudel, Tara Ballav Adhikari, Pratik Khanal, Ramesh Bhatta, Rajan Paudel, Sandesh Bhusal, Prem Basel.

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
