## [Decision Letter · Decision Letter 0]

2 Aug 2021

 PGPH-D-21-00378 Sleep quality and its correlates among undergraduate medical students in Nepal: a cross-sectional study PLOS Global Public Health

Dear Dr. Paudel,

Thank you for submitting your manuscript to PLOS Global Public Health. After careful consideration, we feel that it has merit but does not fully meet PLOS Global Public Health’s publication criteria as it currently stands. Therefore, we invite you to submit a revised version of the manuscript that addresses the points raised during the review process.

 The reviewers have identified major issues in the methodology results and references which needs clarification. Reviewer-2 have suggested to add more details on the rationale of the study. While Plos Global Public Health do not review the manuscripts based on the novelty or innovation, we recommend authors to provide stronger rationale for the study.  

We look forward to receiving your revised manuscript.

Kind regards,

Roopa Shivashankar, MD, MSc

Academic Editor

Journal Requirements:

Additional Editor Comments (if provided):

Reviewers' comments:

Reviewer's Responses to Questions

**Comments to the Author**

1. Does this manuscript meet PLOS Global Public Health’s publication criteria? Is the manuscript technically sound, and do the data support the conclusions? The manuscript must describe methodologically and ethically rigorous research with conclusions that are appropriately drawn based on the data presented.

Reviewer #1: Yes

Reviewer #2: Yes

2. Has the statistical analysis been performed appropriately and rigorously?

Reviewer #1: Yes

Reviewer #2: Yes

3. Have the authors made all data underlying the findings in their manuscript fully available (please refer to the Data Availability Statement at the start of the manuscript PDF file)?

Reviewer #1: Yes

Reviewer #2: Yes

4. Is the manuscript presented in an intelligible fashion and written in standard English?

Reviewer #1: Yes

Reviewer #2: Yes

5. Review Comments to the Author

Reviewer #1: 1. Authors have mentioned that the students having communication difficulties were excluded. However the study participants were chosen randomly from college list, how communication difficulties were assessed?

2. All the study tools were self-administered through internet, what was the reason to exclude students who had communication difficulty.

3. Authors have not explained & mentioned how they have addressed the selection bias. Students having higher grades were selected in the study which could lead to the bias as the possibility of studying more hours and sleeping less (poor quality sleep) is higher in students having higher grades.

4. More studies could be added to the discussion section of the study.

Reviewer #2: 1. Summary

The aim of the study was to determine the prevalence of poor sleep quality and its correlates among medical students in a government medical college in Nepal. The authors report a relatively high prevalence of poor sleep quality (38.2%), and identify a number of factors that correlate with this outcome. The reported prevalence is similar or lower than those reported by other studies. The factors influencing sleep quality are largely those that have been previously investigated by other studies, and except for internet addiction and smoking habits, the results align with the literature.

The strengths of the study include their robust methods in relation to the research question, detailed discussion of their results and the conclusions and recommendations being supported by the data presented.

A major weakness of this manuscript is the introduction in two different ways. First, it fails to lay out the reasons or the precise motivation for undertaking this study, since there is recent available literature from Nepal and other countries on sleep quality in medical students, and well as their correlates. In the absence of specific research gaps being highlighted, it is difficult for a reader to evaluate how this manuscript advances our understanding of this area of research. Second, there are several examples of where the reference provided fails to back up the facts stated. I have provided more details below. A third weakness relates to the quality of language used in this manuscript (examples provided below).

My overall recommendation is therefore to address these two major weaknesses, especially towards highlighting the advance this paper makes, as well as language editing to clarify important points

2. Major issues

Abstract

In conclusion, the authors only highlight depressive symptoms as an important correlate of poor sleep quality, even though other factors are also described in the results. I would recommend to change this to provide a more balanced conclusion based on what the authors report as their results.

Introduction

- The following references do not match the facts stated: 5 (not a WHO report), 7, 8, 9 (in paragraph 3)

- The following statements need clarifications:

o “other studies reporting 50 to 70 million people to be chronically suffering from sleep disorders (6)” - clarify that this is only in the American population

o “Proper sleep aids in the well-functioning of the brain” – reword to “optimum functioning of the brain”

o “improve knowledge and grasp new things” – reword to “grasp new concepts”

o Clarify – “Medical students are prone to developing stressful academics and rigorous medical education”

o Clarify – “Medical students are prone to developing stressful academics and rigorous medical education”

o Clarify – “Some studies revealed that over 70% of the students face some sort of sleep-related problems” – how does this related to the previous statement which provide ranges between 24-49%? Also note that REF #12 is only a preliminary study, hence the authors must exercise caution while referring to the statistics in the introduction.

o Clarify – “In previous studies conducted in Nepal, the prevalence of poor sleep quality was reported to be 44.2% among medical students and 33.4% among undergraduate non-medical students (14,15) – how was the 33.4% derived since the reference doesn’t list this number?

o Clarify – “long duration of the study year” and “Despite their limited health break”

o Given the available literature on the topic as referenced by the authors (also see: https://www.ncbi.nlm.nih.gov/pmc/articles/PMC8261762.1/), can the authors clarify the following statement: “there is a paucity of evidence on the sleep quality among medical students in Nepal.”

Methods

- Clarify – “Participants who had communication difficulty during the study period and below 18 years old.” – how did you define communication difficulty and is this an exclusion criterion? The sentence is incomplete.

- Clarify – “In the excel sheet roll no of the students were kept on one column of each year, and in the next column, a random (rand) command was used, and students with higher scores were selected for participation each year.” – what is meant by higher scores? The method by which the 250 participants were selected from across the 1st-5th year students is not clear.

- The study tools used should be described in a little more detail in terms of their use in global and local settings, whether they are validated for the setting in which they have been used, how was validity checked in case not, and how were the cut-offs identified for categorizing into various categories.

- While analysing data, why was the outcome categorized and logistic regression the only method chosen? Since categorization is known to reduce the power of the analysis, I would recommend the authors present results from both linear and logistic regressions using continuous and categorical data both, to enable a further refinement in our ability to interpret the associations

- Clarify – “The adjusted odds ratio was calculated at a 95% confidence interval (CI)” – language editing?

- Also, a description of the confounding variables used in adjusted analysis is missing, along with a description of why these variables can potentially confound the relationship between exposures and outcomes.

Discussion

- In places where no references are provided (see example below), the authors should state that it is a speculation

- “The country-wise variation could be due to the divergence of sociocultural factors and differences in the teaching-learning environment.”

- In cases of such speculation, it would help for the authors to expand on their thought. For the above example, what specific sociocultural factors and differences do the authors have in mind that could lead to differences in sleep quality among medical students?

- This is applicable to the rest of the discussion as well, whenever references are not provided.

- In comparing sleep quality between first and fourth year students, the authors make a statement of higher academic and clinical load on the fourth year students. What happens to 5th year students? What are the protective factors for the 5th year students that bring down the odds from 3.6 in 4th year to 1.8 in 5th year students?

- What are the authors thoughts on why smartphone addiction was not associated with sleep quality in this sample, as against what the literature states as the authors rightly point out?

Writing quality & clarity

Recommend language editing throughout, especially in statements highlighted.

6. PLOS authors have the option to publish the peer review history of their article (what does this mean?). If published, this will include your full peer review and any attached files.

**Do you want your identity to be public for this peer review?** For information about this choice, including consent withdrawal, please see our Privacy Policy.

Reviewer #1: No

Reviewer #2: **Yes: **Debarati Mukherjee

---

## [Editor Report · Decision Letter 1]

26 Oct 2021

PGPH-D-21-00378R1

Sleep quality and its correlates among undergraduate medical students in Nepal: a cross-sectional study

Dear Dr. Paudel,

Thank you for submitting your manuscript to PLOS Global Public Health. After careful consideration, we feel that it has merit but does not fully meet PLOS Global Public Health’s publication criteria as it currently stands. Therefore, we invite you to submit a revised version of the manuscript that addresses the points raised during the review process.

1. Add the possible selection bias that may have occurred due to non-response in the limitations. 

2. While you have responded that the tools have been used in Nepal in the past, it is unclear whether tools were validated in the language and population. Please add details regarding validation. 

3. Describe how each of the confounding factors information was collected in the methods section. 

We look forward to receiving your revised manuscript.

Kind regards,

Roopa Shivashankar, MD, MSc

Academic Editor
---

## [Editor Report · Decision Letter 2]

9 Nov 2021

PGPH-D-21-00378R2

Sleep quality and its correlates among undergraduate medical students in Nepal: a cross-sectional study

Dear Dr. Paudel,

Thank you for submitting your manuscript to PLOS Global Public Health. After careful consideration, we feel that it has merit but does not fully meet PLOS Global Public Health’s publication criteria as it currently stands. Therefore, we invite you to submit a revised version of the manuscript that addresses the points raised during the review process.

1.      Add the possible selection bias that may have occurred due to non-response in the limitations. 

Thank you editor for suggesting us to write there might be possible selection bias that may have occurred due to non-response in the limitation. We acknowledge the editor’s comments and added on it on our manuscript as:

“*There might be selection bias that may have occurred due to non-response*”.

**Editors comment: Please explain how the possible selection bias would have affected the study results. That is,  if the non-responders are likely to have better or worse sleep quality than responders, how does that affect the study results. **

2.      Describe how each of the confounding factors information was collected in the methods section. 

We have described on methods section; how the confounding factors information was collected as:

“In bi-variable logistic analysis, variables namely depression, smart mobile phone addiction, current alcohol intake status, current smoking status, study year, study choice, academic achievement, and self-reported health problems were found to have p-value less than 0.1. These variables fulfilled minimum requirements for further multivariable logistic regression and was thus fitted in the final regression model. Multivariable logistic regression analysis was conducted to determine the statistically significant association between explanatory variables and outcome variables. The adjusted odds ratio was calculated at a 95% confidence interval (CI), and a p-value less than 0.05 was considered statistically significant.”

**Editor comments: In the methods (not in the analysis part), explain how the variables (**
**depression, smart mobile phone addiction, current alcohol intake status, current smoking status, study year, study choice, academic achievement, and self-reported health problems) were collected. That is, explain how the depression, smartphone addiction or current alcohol use were ascertained or defined in the study. **

**Additional Comments**

Check the language, grammer and typos throughout the manuscript.Add reference to the sentence (page 3, second para)A higher prevalence of poor sleep quality among medical students than non-medical students and general population has also been reported.   

Incomplete sentence. Rephrase. (Page 4, para 1)

Despite their limited health break due to inadequate vacation and long duration of hospital posting coupled with academic sessions and stressful lifestyle.

Please edit the sentence as below. Also explain the why the students at public funded medical college will be different from the private funded college. (Page 4, para 1)

Our study in a public-funded medical college is first of its kind as the previous published studies were conducted in private medical colleges.

We look forward to receiving your revised manuscript.

Kind regards,

Roopa Shivashankar, MD, MSc

Academic Editor
---

## [Editor Report · Decision Letter 3]

7 Dec 2021

Sleep quality and its correlates among undergraduate medical students in Nepal: a cross-sectional study

PGPH-D-21-00378R3

Dear Dr. Paudel,

We're pleased to inform you that your manuscript has been judged scientifically suitable for publication and will be formally accepted for publication once it meets all outstanding technical requirements.

Within one week, you'll receive an e-mail detailing the required amendments. When these have been addressed, you'll receive a formal acceptance letter and your manuscript will be scheduled for publication.

An invoice for payment will follow shortly after the formal acceptance. To ensure an efficient process, please log into Editorial Manager at https://www.editorialmanager.com/pgph/ click the 'Update My Information' link at the top of the page, and double check that your user information is up-to-date. If you have any billing related questions, please contact our Author Billing department directly at authorbilling@plos.org.

Kind regards,

Roopa Shivashankar, MD, MSc

Academic Editor